# *Aspergillus fumigatus* Protease Alkaline Protease 1 (Alp1): A New Therapeutic Target for Fungal Asthma

**DOI:** 10.3390/jof6020088

**Published:** 2020-06-16

**Authors:** Kirk M. Druey, Morgan McCullough, Ramaswamy Krishnan

**Affiliations:** 1Lung and Vascular Inflammation Section, Laboratory of Allergic Diseases, National Institute of Allergy and Infectious Diseases/National Institutes of Health, Bethesda, MD 20892, USA; morgan.mccullough@nih.gov; 2Center for Vascular Biology Research, Department of Emergency Medicine, Beth Israel Deaconess Medical Center, Boston, MA 02215, USA; rkrishn2@bidmc.harvard.edu

**Keywords:** *Aspergillus fumigatus*, airway smooth muscle, contraction, AHR

## Abstract

We review three recent findings that have fundamentally altered our understanding of causative mechanisms underlying fungal-related asthma. These mechanisms may be partially independent of host inflammatory processes but are strongly dependent upon the actions of Alp1 on lung structural cells. They entail (i) bronchial epithelial sensing of Alp1; (ii) Alp1-induced airway smooth muscle (ASM) contraction; (iii) Alp1-induced airflow obstruction. Collectively, these mechanisms point to Alp1 as a new target for intervention in fungal asthma.

## 1. Introduction

Asthma is a widely prevalent and incurable respiratory disease that affects nearly 300 million people worldwide [1]. The cardinal feature of asthma is exaggerated airflow obstruction in response to numerous contraction triggers present in inflamed lungs—a condition termed “airway hyper-responsiveness” (AHR). Although the clinical heterogeneity of asthma was recognized decades ago, only recently have we appreciated the existence of unique molecular pathways leading to AHR—termed “endotypes”—that underlie disease in distinct subpopulations [2]. One such endotype, comprising nearly 15–30 million patients, is termed “severe asthma with fungal sensitization” (SAFS) or simply fungal asthma [3,4,5,6]. These patients are sensitized (i.e., have IgE-mediated allergy) to common molds, in particular *Aspergillus fumigatus* (*Af*).

Fungal asthma patients frequently experience persistent and life-threatening symptoms of airflow obstruction despite guideline-based treatments with high-dose inhaled corticosteroids plus β-agonist bronchodilators. Additional treatment is largely focused on alleviating endobronchial fungal infection or airway inflammation. Whether this is attempted using anti-fungal agents or neutralizing antibodies against specific cytokines or mediators (e.g., anti-IgE, anti-IL-4/5/13) [7], clinical success has been inconsistent [8,9]. It is logical to conclude that there are fundamental mechanisms at play that are insensitive to a reduction in airway fungal burden or allergic inflammation.

Here, we review recent studies supporting the hypothesis that fungal allergens exert direct effects on lung structural cells independent of the inflammatory response and discuss the possibility that specific targeting of allergen protease activity represents a viable strategy for the treatment of AHR in fungal asthma.

## 2. Role of Fungal Proteases in Initiation of Allergic Airway Inflammation

### 2.1. A Fungal Protease Allergen Infiltrates the Bronchial Submucosa in Asthma

A critical gatekeeper of innate lung immunity is the ciliated bronchial epithelium. Under normal circumstances, it provides a physical barrier to allergens and actively secretes mucus to clear allergens [10]. These functions are impaired by chronic allergen exposure [11,12], and emerging evidence suggests that such impairment can be induced by allergen proteases, including those from fungi. *Af*-derived serine protease activity may simultaneously activate epithelial cells and disrupt barrier integrity. In one study, *Af*-secreted filtrates or conidia, when applied to respiratory epithelial cells in vitro, elicited epithelial detachment and IL-6 and IL-8 secretion [13]. In a separate study, treatment of epithelial cells with *Af* spores in vitro induced formation of F-actin stress fibers (detected by phalloidin staining) and disruption of multi-protein force-bearing structures which cells use to attach to extracellular matrix (ECM) substrates called focal adhesions (FAs) (assessed by immunostaining for the FA component vinculin) [14]. In each case, inhibition of serine protease activity reversed the deleterious effects of *Af*. Beyond *Af*, application of an alkaline serine protease allergen from *Penicillium chrysogenum* (*Pen ch 13*) to respiratory cells in vitro led to degradation of the epithelial tight junction protein occludin, while concurrently promoting IL-8 and PGE2 secretion [15]. In another study, treatment of epithelial cells with *Per a 10* serine protease from *Periplaneta Americana* (cockroach) in vitro increased the permeability of monolayers through the degradation of ZO-1 and occludin [16]. Taken together, these findings suggest that allergen serine proteases may induce epithelial barrier dysfunction in asthma through cytoskeletal rearrangements, FA disruption, and breakdown of cell–cell contacts.

Within the respiratory epithelium and bronchial submucosa of lung biopsies from subjects with asthma, we identified immunoreactive Alp1, a known serine protease allergen from *Af* (Asp f 13) (Figure 1a) [17]. Submucosal Alp1 co-localized with ASM cells, suggesting potentially pathogenic interactions with ASM. Likewise, we observed increased Alp1 immunoreactivity within the ASM bundle in mice sensitized with *Af* filtrates and challenged intranasally with an allergen compared to saline-challenged controls [18]. Alp1 is a member of the subtilisin/peptidase S8 family of serine proteases, which can be found in bacteria, fungi, and archaea species [19]. These proteases contain a characteristic Asp/His/Ser catalytic triad but are not substrate-specific [20]. The active site is formed by tertiary conformation of the protease, where the Ser residue initiates a nucleophilic attack on the carbonyl carbon atom of the substrate. The His residue accepts a proton from the Ser residue, resulting in cleavage of the substrate at the amino terminus [21].

### 2.2. Mechanisms Underlying Allergen Protease Sensing and Tissue Deposition

The mechanism(s) underlying Alp1 deposition in the bronchial submucosa in asthma remain unclear. In a recent study, co-administration of Alp1 and biotin to the airways of naïve mice elicited widespread disruption of airway epithelial integrity, as evidenced by the presence of biotin in lung tissue [20]. Alp1 also reduced the transepithelial resistance (TER) of epithelial cell monolayers in vitro and degraded the adherens junction protein E-cadherin, indicating increased para-cellular permeability. However, we did not detect Alp1 deposits within ASM bundles of lung biopsies from naïve mice exposed to *Af*-secreted filtrates for short periods of time in the absence of allergic airway inflammation, suggesting that Alp1 protease activity alone is not sufficient for tissue deposition [18]. On the other hand, mice sensitized and challenged with heat-inactivated *Af* filtrates, which lack protease activity, exhibited allergic airway inflammation and AHR, yet no Alp1 was detected in the bronchial submucosa in these mice. [18]. Taken together, these results suggest that both Alp1 protease activity and allergic airway inflammation may be necessary for the accumulation of fungal protease in ASM-containing areas of the lung.

Respiratory epithelial cells can actively sense Alp1, which may facilitate permeation into lung tissue. Wiesner et al. proposed that the voltage-gated Ca^2+^ channel TRPV4 is a mechanosensor of Alp1-induced junctional damage in epithelial cells. Activation of TRPV4 by Alp1 elicits a Ca^2+^-calcineurin-mediated allergic inflammatory cascade initiated by bronchiolar club cells (specialized cells within the ciliated epithelium) and recruitment of monocyte-derived dendritic cells (DCs) and T helper type 2 (Th2) cells [20]. Club-cell-specific Trpv4 deletion in mice resulted in significantly decreased, but not absent, DC, eosinophil and Th2 recruitment to the lungs. However, whether or not TRPV4 interacts directly with Alp1 or is required for allergen infiltration into bronchial submucosa was not investigated.

Another candidate sensor of Alp1 on airway epithelial cells is the protease-activated receptor 2 (PAR2). PAR2 expression is required for the development of allergic inflammation induced by subtilisin [22]. PAR2 is also known to be activated by allergen proteases including Der p 9 and Der p 3 [23] from house dust mites, as well as the alkaline serine protease (AASP) of the mold *Alternaria alternata*, and Pen c 13 from the fungus *P. citrinum* [24,25,26]. In contrast, PAR2 peptides are not cleaved by Alp1 [20]. Furthermore, PAR2 inhibition does not affect Alp1-induced mediator (PGE2) secretion by human bronchial epithelial cells [27]. Thus, any role for PAR2 or other PARs in the epithelial response to Alp1 is unclear.

*Af* may also co-opt cellular machinery to promote active secretion of protease into the bronchial submucosa. Fernandes et al. describe a mechanism whereby *Af* hyphae can traverse bronchial epithelial cells by recruiting cellular actin and forming a “tunnel” [28]. This process does not overtly perturb epithelial integrity, raising the possibility that viable fungus present in the airways may secrete protease directly into the bronchial submucosa independent of inflammatory mechanisms.

### 2.3. Alp1 as a Potential Biomarker of Fungal Asthma

Alp1 immunoreactivity was most prominent in lungs from patients with severe disease while minimal in biopsies from healthy controls (Figure 1a and Table 1). The extent of Alp1 tissue coverage correlated with lung functional impairment including reduced forced expiratory volume in one second (FEV_1_) (Figure 1b) and the provocative concentration of the bronchoconstrictor methacholine (MCh) needed to reduce FEV1 by 20% (PC_20_FEV_1_), two standard clinical measurements used to estimate the severity of AHR [29]. Quantities of culturable *Af* from sputum do not correlate well with reduced lung function in asthma [30]. As we have detected Alp1 in sputum from patients with fungal asthma [17] and in bronchoalveolar lavage fluid from *Af*-challenged mice [18], quantification of Alp1 levels in respiratory fluids could provide an alternative diagnostic tool for patient stratification in severe asthma. Analysis of sputum protease activity following treatment with Alp1 inhibitors might serve as a noninvasive indicator of response to therapy.

## 3. Alp1 Induces Airway Smooth Muscle Contraction

Even in the absence of exogenous spasmogens, Alp1 elicits contraction of cultured human ASM cells (Figure 2a) [18,31]. Alp1 also enhances bronchoconstriction in MCh-treated precision-cut lung slices (PCLS) from naïve mice and healthy human controls [18] (Figure 2b). These phenotypes require Alp1 protease activity but not pre-established allergic inflammation [31].

Alp1 enhances ASM force generation through several potentially overlapping mechanisms [18]. Alp1 degrades the ECM, which perturbs integrin-mediated ECM-ASM attachments and increases activity of the GTPase RhoA in ASM. A crucial effector of activated RhoA, the Rho-associated coiled-coil forming kinase (ROCK), thereby inhibits the activity of myosin light chain phosphatase MYPT1, which in turn increases myosin light chain 2 (MLC2) phosphorylation. In total, these steps facilitate myosin–actin filament interactions, cellular shortening, and contraction. In accordance with this cascade, inhibiting ROCK abolished Alp1-induced ASM hypercontraction [31]. Beyond this cascade, Alp1 also enhances intracellular Ca^2+^ mobilization in response to contractile agonists [18], although the mechanisms underlying this phenotype are less clear. For example, Alp1-induced Ca^2+^ mobilization in ASM cells could involve TRPV4, as was recently reported to be the case for bronchiolar club cells [20], but this remains to be ascertained.

RhoA can induce ASM hypercontraction through additional mechanisms. One such mechanism entails recruitment and activation of signaling modules containing paxillin, vinculin, and focal adhesion kinase (FAK) at focal adhesions, complexes which ultimately induce actin polymerization [32,33]. The extent to which RhoA-mediated actin polymerization, focal adhesion and cytoskeletal re-organization, and perturbations to cell–cell physical interactions contributes to Alp1-induced ASM hypercontraction needs further investigation. Answering this question will require multiplexed measurements of mechanotransduction, ASM force generation, and ASM force transmission using the methods of contractile force screening in cultured ASM cells [34,35] and tissue traction microscopy in PCLS [36].

## 4. Alp1 Induces Inflammation and AHR In Vivo

In patients with asthma, AHR is defined by exaggerated airflow limitation in response to inhalation of a bronchoconstrictor (typically MCh). In mice, AHR is evaluated using invasive plethysmography, a technique that allows direct quantification of airway resistance following administration of inhaled MCh [37]. Allergen-associated airway remodeling, bronchoconstriction, and increased mucus in the lumen of conducting airways all contribute to murine AHR [38]. Serine proteases induce inflammation and AHR in vivo. Subtilisin, which is a component of commercial detergents, induces type 2 lung inflammation in mouse models and has been linked to occupational asthma in humans [22]. Fungal serine proteases *Pen c 13* from *Penicillium citrinum* and *Epi p 1* from *Epicoccum purprascens* also induce allergic airway inflammation in mice [39,40]. Likewise, repetitive challenge of mice with purified or recombinant Alp1 to the lower respiratory tract induces allergic inflammation including eosinophil recruitment to the airways, tissue remodeling (mucus hypersecretion), and increased levels of type 2 cytokines (eg IL-4, IL-5, IL-13) [20,41] as well as AHR [31,42]. In our experiments, Alp1-challenged wild-type mice (Balb/c or C57Bl/6 strains) spontaneously developed increased baseline lung resistance compared to naïve controls, a finding that mirrors the spontaneous airflow limitation that is a prominent feature of asthma in humans [31]. The predisposition to develop AHR was independent of eosinophils and PAR2 [31]. By contrast, Wiesner et al. demonstrated a requirement for both innate lymphoid type 2 cells (ILC2s) and T lymphocytes in Alp1-induced allergic airway inflammation [20].

It is unknown whether lymphocyte-deficient mice develop AHR following Alp1 challenge or whether Alp1 accumulates in lung tissue in the absence of type 2 inflammation. Assessment of bronchoconstriction in immune-deficient mouse strains and identification of “hotspots” of Ca^2+^ signaling, RhoA activity, and/or other perturbations of contraction-inducing pathways in Alp1-containing lung tissue will be needed to determine the extent to which the protease elicits AHR through inflammation-independent mechanisms.

## 5. Conclusions and Future Directions

In this review, we describe recent discoveries that (i) confirm the long-held assumption that ASM hypercontraction is a functionally dominant derangement in severe asthma [43], and, (ii) suggest a unique ASM cytoskeletal mediator, Alp1, whose effectors appear to act partially independently of contraction- or relaxation-inducing GPCRs [18] but depend on ASM–ECM interactions.

A critical therapeutic gap is to identify moieties with specific affinity for fungal Alp1 but without activity on endogenous serine proteases present in the human airway. One appealing possibility is that these moieties can be identified through high-throughput screening (HTS) (Figure 3). First, existing libraries of commercially available protease inhibitors can be screened to detect inhibition of Alp1 protease activity on a fluorescent substrate (casein). Next, using the methods of contractile force screening in cultured human ASM cells [34,35], and tissue traction microscopy in human PCLS [36], primary hits can be rank-ordered based on inhibition of Alp1-induced ASM hypercontraction. A prioritized set of efficacious and non-toxic hits can be evaluated for its effects on (i) ASM cells from subjects with fungal asthma; (ii) airways of murine and human PCLS treated with Alp1 [18]; (iii) murine models of SAFS [31]. We expect these studies to identify novel anti-SAFS drug candidates as well as provide novel insight into both ASM-intrinsic and allergen-protease-dependent mechanisms of bronchoconstriction.

The prevalence of allergy to multiple fungi may be as high as 25% in people with allergy to at least one mold [44,45]. As noted above, Alp1 is highly homologous to serine proteases from a wide variety of allergenic fungi, including *Stachybotrys chartarum* (49%), *Fusarium* (55%), *Penicillium, Candida,* and *Trichophyton* spp. (>40%), including characteristic catalytic residues. Inhibitors of Alp1 protease may thus provide clinical benefit to patients with severe asthma who are sensitized to many common molds.

## Figures and Tables

**Figure 1 jof-06-00088-f001:**
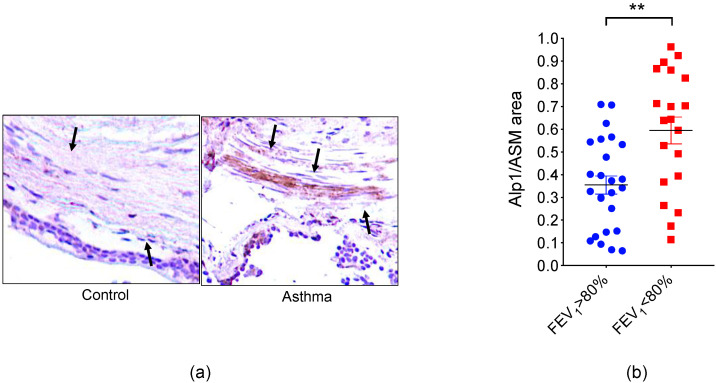
Alp1 co-localizes with bronchial smooth muscle in asthma. (**a**) Representative Alp1 immunohistochemistry in lung biopsies from a cohort of healthy control subjects (FEV1 > 80% predicted) and patients with asthma (FEV1 < 80% predicted). Patient demographics are listed in Table 1. Arrows indicate bronchial smooth muscle [17]. (**b**) Quantification of Alp1 in bronchial smooth muscle bundles correlated with impairment of lung function; ** *p* = 0.003, Mann–Whitney.

**Figure 2 jof-06-00088-f002:**
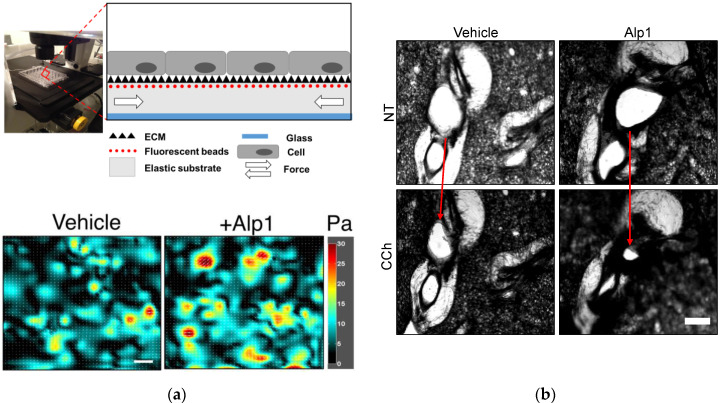
Alp1 induces ASM contraction. (**a**) Human ASM cells were cultured to confluence upon elastic polydimethylsiloxane (PDMS) NuSil^®^ Gel-8100 (NuSil Silicone Technologies, CA) substrates prepared in 96-well plates as outlined in [35]. As the cells contract, they displace the substrate; based on the displacement of substrate-bound fluorescent beads relative to a cell-free image, and with knowledge of substrate stiffness and thickness, cellular contractile maps can be obtained. Such maps were obtained for ASM cells treated with either vehicle (average value of ASM contraction from map = 8.1 Pa) or Alp1 (average value of ASM contraction from map = 10 Pa). Scale bar = 100 µm. (**b**) Airway constriction (arrows) in human PCLS pre-treated with Alp1 or vehicle and stimulated with carbachol (CCh) as outlined in [18].

**Figure 3 jof-06-00088-f003:**
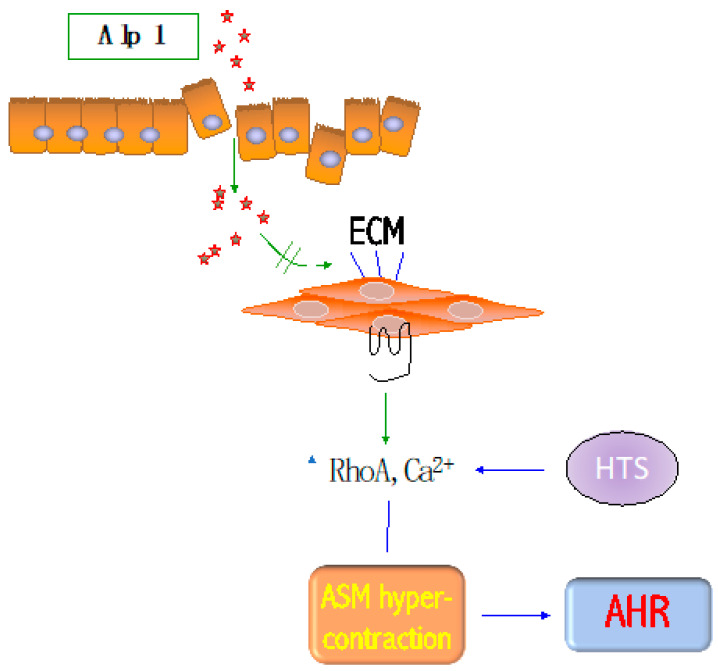
Unique mechanism to be exploited for fungal asthma treatment. Alp1 infiltrates the bronchial mucosa and provokes bronchoconstriction by degrading the extracellular matrix (ECM). This results in increased RhoA activation and Ca^2+^ signaling to GPCR agonists, leading to ASM hypercontraction and AHR. Inhibitors of Alp1-induced ASM hypercontraction may be identified by high-throughput screening (HTS) to treat and reveal underlying mechanisms responsible for AHR.

**Table 1 jof-06-00088-t001:** Demographics of asthma cohort.

	FEV1 > 80%	FEV1 < 80%
**No.**	17	18
**Age ((mean, range,yr.) ±.) ***	32.7 (18−56)	46 (23−68)
**Sex (M/F)**	8/9	7/11
***Af* sensitivity ****	9/17	7/18
**Inhaled CCS ^#&^**	958.3	1656

* *p* = 0.005, ^&^
*p* = 0.04, Mann–Whitney; ** Positive skin prick test to *Af*; ^#^ range equivalent of fluticasone in micrograms/day.

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
