# Peer review of "Aspergillus fumigatus Protease Alkaline Protease 1 (Alp1): A New Therapeutic Target for Fungal Asthma"

_jof, 2020, doi:10.3390/jof6020088_

Round 1

Reviewer 1 Report

The authors review the current knowledge of Aspergillus fumigatus protease alkaline protease 1 (Alp1), and suggest that as a new therapeutic target for fungal asthma. 

The review nicely summarises the recent findings in mouse and man, and the authors have provided also additional data related to previously published studies. The review would have given opportunities to take a step forward, and the manuscript could get deeper, if you:

-compare in more details the properties of Alp1 to other known enzyme allergens e.g. sequence, structure, mechanism and associated hypotheses  

-make a schematic presentation how A. fumigatus leads to asthma and what kind of effects it has in the ASM.  

-discuss more broadly and suggest plans, how to study and develop new possible therapeutic approaches in the future.

Also, please provide at least the basic demographic information of the patients (age, sex, do they have specific IgEs, are they responsible to corticosteroids etc), whose FEVs you show in Fig. 1b.

Author Response

Response to Reviewer 1 Comments

The authors review the current knowledge of Aspergillus fumigatus protease alkaline protease 1 (Alp1), and suggest that as a new therapeutic target for fungal asthma. 

Point 1: The review nicely summarises the recent findings in mouse and man, and the authors have provided also additional data related to previously published studies. The review would have given opportunities to take a step forward, and the manuscript could get deeper, if you:

-compare in more details the properties of Alp1 to other known enzyme allergens e.g. sequence, structure, mechanism and associated hypotheses  

Response 1: We have integrated this information throughout the paper as follows:

(Section 2.1, lines 54-69):

Af-derived serine protease activity may simultaneously activate epithelial cells and disrupt barrier integrity.  In one study, Af secreted filtrates or conidia when applied to respiratory epithelial cells in vitro elicited epithelial detachment and IL-6 and IL-8 secretion [13]. In a separate study, treatment of epithelial cells with Af spores in vitro induced formation of F-actin stress fibers (detected by phalloidin staining) and disruption of multi-protein force-bearing structures which cells use to attach to extracellular matrix (ECM) substrates called focal adhesions (FAs) (assessed by immunostaining for the FA component vinculin) [14]. In each case, inhibition of serine protease activity reversed the deleterious effects of Af. Beyond Af, application of an alkaline serine protease allergen from Penicillium chrysogenum (Pen ch 13) to respiratory cells in vitro led to degradation of the epithelial tight junction protein occludin while concurrently promoting IL-8 and PGE2 secretion [15]. In an another study, treatment of epithelial cells with Per a 10 serine protease from Periplaneta Americana (cockroach) in vitro increased permeability of monolayers through the degradation of ZO-1 and occludin [16].

(Section 2.1, lines 79-85):

Alp1 is a member of the subtilisin/peptidase S8 family of serine proteases that can be found in bacteria, fungi, and archaea species [48]. These proteases contain a characteristic Asp/His/Ser catalytic triad but are not substrate-specific [19]. The active site is formed by tertiary conformation of the protease, where the Ser residue initiates nucleophilic attack on the carbonyl carbon atom of the substrate. The His residue accepts a proton from the Ser residue, resulting in cleavage of the substrate at the amino terminus [20].

(Section 2.2, lines 113-121):

Another candidate sensor of Alp1 on airway epithelial cells is the protease activated receptor 2 (PAR2). PAR 2 expression is required for the development of allergic inflammation induced by subtilisin [21]. PAR2 is also known to be activated by allergen proteases including Der p 9 and Der p 3 [22] from house dust mites, as well as the alkaline serine protease (AASP) of the mold Alternaria alternata, and Pen c 13 from the fungus P. citrinum [23]. [24,25]. In contrast, PAR2 peptides are not cleaved by Alp1 [19]. Furthermore, PAR2 inhibition does not affectthat Alp1-induced mediator (PGE2) secretion by human bronchial epithelial cells [26]. Thus, any role for PAR2 or other PARs in the epithelial response to Alp1 is unclear.

(Section 4, lines 179-184):

Serine proteases induce inflammation and AHR in vivo. Subtilisin, which is a component of commercial detergents, induces type 2 lung inflammation in mouse models and has been linked to occupational asthma in humans [21]. Fungal serine proteases Pen c 13 from Penicillium citrinum and Epi p 1 from Epicoccum purprascens also induce allergic airway inflammation in mice [38,39].

Point 2: -make a schematic presentation how A. fumigatus leads to asthma and what kind of effects it has in the ASM.

Response 2: Done (see new Figure 3).

Point 3: -discuss more broadly and suggest plans, how to study and develop new possible therapeutic approaches in the future.

Response 3: Done (see Section 5, lines 213-224):

One appealing possibility is that these moieties can be identified through high-throughput screening (HTS) (Figure 3). First, existing libraries of commercially available protease inhibitors can be screened to detect inhibition of Alp1 protease activity on a fluorescent substrate (casein). Next, using the methods of contractile force screening in cultured human ASM cells [33,34], and tissue traction microscopy in human PCLS [35], primary hits can be rank-ordered based on inhibition of Alp1-induced ASM hypercontraction. A prioritized set of efficacious and non-toxic hits can be evaluated for its effects on (i) ASM cells from subjects with fungal asthma; (ii) airways of murine and human PCLS treated with Alp1 [18]; (iii) murine models of SAFS [30]. We expect these studies to identify novel anti-SAFS drug candidates as well as provide novel insight into both ASM-intrinsic and allergen protease-dependent mechanisms of bronchoconstriction.

Point 4: Also, please provide at least the basic demographic information of the patients (age, sex, do they have specific IgEs, are they responsible to corticosteroids etc), whose FEVs you show in Fig. 1b.

Response 4: Available information on the subjects is now shown in new Table 1.

Reviewer 2 Report

This is a well balanced review summarising the key studies that have dissected how a specific protease secreted by Aspergillus fumigatus (Af), alkaline protease 1 (Alp1), mediates fungal-related asthma. I only have minor suggestions.

A schematic figure summarising the key themes of the review, showing the model of how Alp1 mediates fungal asthma would be welcome. Indicating on the figure which ones could be therapeutically targeted (section 5) would also significantly add to the review.

The authors abbreviation of Aspergillus fumigatus to Af. Yet there are instances when it is not in italics (e.g. line 77).

The authors discuss studies that have assessed the role of Af spores in mediating epithelial cell response and that these responses can be reversed by inhibition of serine protease activity (section 2.1). However, unlike in other parts of the review the precise assays undertaken to show these results were not taken (especially when discussing refs 13-15). My understanding is that these were from in vitro epithelial cell assays? This an important clarification.

Author Response

Response to Reviewer 2 Comments

This is a well balanced review summarising the key studies that have dissected how a specific protease secreted by Aspergillus fumigatus (Af), alkaline protease 1 (Alp1), mediates fungal-related asthma. I only have minor suggestions.

Point 1: A schematic figure summarising the key themes of the review, showing the model of how Alp1 mediates fungal asthma would be welcome. Indicating on the figure which ones could be therapeutically targeted (section 5) would also significantly add to the review.

Response 1: Done (see new Figure 3).

Point 2: The authors abbreviation of Aspergillus fumigatus to Af. Yet there are instances when it is not in italics (e.g. line 77).

Response 2: Corrected throughout.

Point 3: The authors discuss studies that have assessed the role of Af spores in mediating epithelial cell response and that these responses can be reversed by inhibition of serine protease activity (section 2.1). However, unlike in other parts of the review the precise assays undertaken to show these results were not taken (especially when discussing refs 13-15). My understanding is that these were from in vitro epithelial cell assays? This an important clarification.

Response 3: Clarified as follows (see Section 2.1, lines 54-69):

Af-derived serine protease activity may simultaneously activate epithelial cells and disrupt barrier integrity.  In one study, Af secreted filtrates or conidia when applied to respiratory epithelial cells in vitro elicited epithelial detachment and IL-6 and IL-8 secretion [13]. In a separate study, treatment of epithelial cells with Af spores in vitro induced formation of F-actin stress fibers (detected by phalloidin staining) and disruption of multi-protein force-bearing structures which cells use to attach to extracellular matrix (ECM) substrates called focal adhesions (FAs) (assessed by immunostaining for the FA component vinculin) [14]. In each case, inhibition of serine protease activity reversed the deleterious effects of Af. Beyond Af, application of an alkaline serine protease allergen from Penicillium chrysogenum (Pen ch 13) to respiratory cells in vitro led to degradation of the epithelial tight junction protein occludin while concurrently promoting IL-8 and PGE2 secretion [15]. In an another study, treatment of epithelial cells with Per a 10 serine protease from Periplaneta Americana (cockroach) in vitro increased permeability of monolayers through the degradation of ZO-1 and occludin [16].

We look forward to your critical appraisal of the revised manuscript and thank you for your consideration.

Sincerely,

Kirk M. Druey, M.D.

Round 2

Reviewer 1 Report

All my comments have been taken into account.

Just a minor notion, at the page 3, line 81, there's a reference 48, which doesn't exist in the reference list and anyway the order number is wrong. Please correct!

Author Response

Corrected